# PiFold: Toward effective and efficient protein inverse folding

**Zhangyang Gao[†], Cheng Tan[†], Stan Z. Li[\*]**
AI Lab, Research Center for Industries of the Future, Westlake University
{gaozhangyang, tancheng,Stan.ZQ.Li}@westlake.edu.cn

## Abstract

How can we design protein sequences folding into the desired structures effectively and efficiently? AI methods for structure-based protein design have attracted increasing attention in recent years; however, few methods can simultaneously improve the accuracy and efficiency due to the lack of expressive features and autoregressive sequence decoder. To address these issues, we propose PiFold, which contains a novel residue featurizer and PiGNN layers to generate protein sequences in a one-shot way with improved recovery. Experiments show that PiFold could achieve 51.66% recovery on CATH 4.2, while the inference speed is 70 times faster than the autoregressive competitors. In addition, PiFold achieves 58.72% and 60.42% recovery scores on TS50 and TS500, respectively. We conduct comprehensive ablation studies to reveal the role of different types of protein features and model designs, inspiring further simplification and improvement. The PyTorch code is available at GitHub.

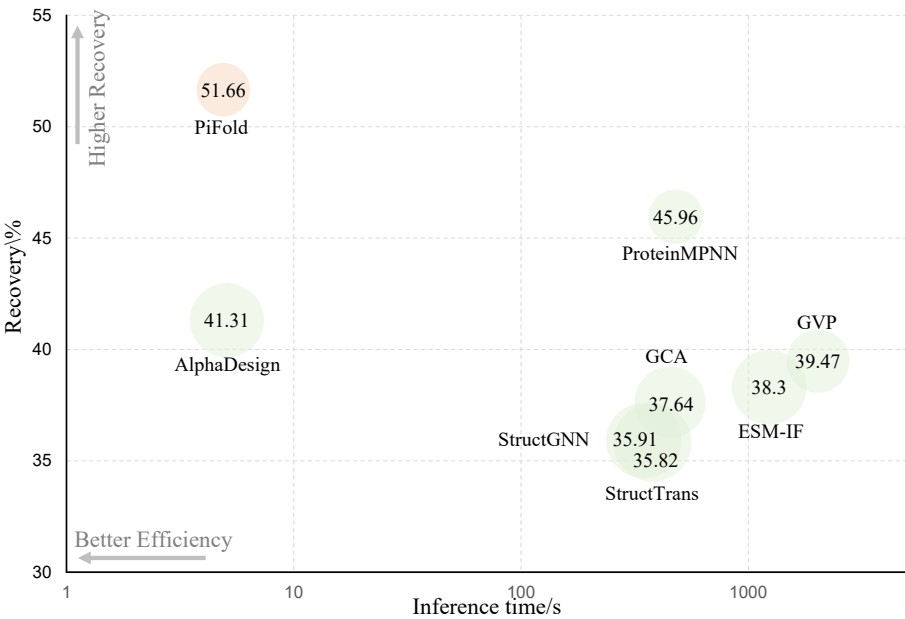

**Figure 1:** Performance comparison with other graph-based protein design methods. The recovery scores, the inference time costs, and the perplexities are shown in the Y-axis direction, the X-axis direction, and the circle size, respectively. Note that the recovery and perplexity results in the CATH dataset are reported without using any other training data. The inference time is averaged over 100 long protein sequences of average length 1632 on an NVIDIA V100.

## 1 Introduction

Proteins are linear chains of amino acids that fold into 3D structures to control cellular processes, such as transcription, translation, signaling, and cell cycle control. Creating novel proteins for human purposes could deepen our understanding of living systems and facilitate the fight against the

---

[†]Equal Contribution, [\*]Corresponding Author.

disease. One of the crucial problems is to design protein sequences that fold to desired structures, namely structure-based protein design (Pabo, 1983). Recently, many deep learning models have been proposed to solve this problem (Li et al., 2014; Wu et al., 2021; Pearce & Zhang, 2021; Ovchinnikov & Huang, 2021; Ding et al., 2022; Gao et al., 2020; 2022a; Dauparas et al., 2022; Ingraham et al., 2019; Jing et al., 2020; Tan et al., 2022c; Hsu et al., 2022; O'Connell et al., 2018; Wang et al., 2018; Qi & Zhang, 2020; Strokach et al., 2020; Chen et al., 2019; Zhang et al., 2020a; Huang et al., 2017; Anand et al., 2022; Strokach & Kim, 2022; Li & Koehl, 2014; Greener et al., 2018; Karimi et al., 2020; Anishchenko et al., 2021; Cao et al., 2021; Liu et al., 2022; McPartlon et al., 2022; Huang et al., 2022; Dumortier et al., 2022; Li et al., 2022a; Maguire et al., 2021; Li et al., 2022b), among which graph-based models have made significant progress. However, there is still room to improve the model's accuracy and efficiency. For example, most graph models could not achieve 50+% sequence recovery in CATH dataset due to the lack of expressive residue representations. Moreover, they also suffer from the autoregressive generation, resulting in low inference speed. We aim to simultaneously improve the accuracy and efficiency with a simple model containing as few redundancies as possible.

For years, graph-based models have struggled to learn expressive residue representations through better feature engineering, more elaborate models, and a larger training dataset. For example, AlphaDesign (Gao et al., 2022a) and ProteinMPNN (Dauparas et al., 2022) point out that additional angle and distance features could significantly improve the representation quality. GraphTrans (Ingraham et al., 2019), GVP (Jing et al., 2020), and GCA (Tan et al., 2022c) introduce new modules considering graph-based message passing, equivariant vectors, and global attention to learn geometric representations from residue interactions. ESM-IF (Hsu et al., 2022) incorporates additional training data to capture more structural learning bias. Although they have made significant progress, two problems remain to be solved under the same data set setting: (1) Is there a better way to construct effective features to facilitate learning residual representations? (2) How can we improve the model to allow it to learn better representation from residue interactions?

Most graph models (Dauparas et al., 2022; Ingraham et al., 2019; Jing et al., 2020; Hsu et al., 2022; Tan et al., 2022c; Hsu et al., 2022) adopt the autoregressive decoding scheme to generate amino acids, dramatically slowing down the inference process. Interestingly, few studies have attempted to improve the model efficiency, perhaps because the efficiency gain requires sacrificing some accuracy (Bahdanau et al., 2015; Vaswani et al., 2017; Ghazvininejad et al., 2019; Geng et al., 2021; Xie et al., 2020; Wang et al., 2019; Gu et al., 2018), while the latter is more important than efficiency in protein design. To address this dilemma, AlphaDesign (Gao et al., 2022a) proposes a parallel self-correcting module to speed up inference while almost maintaining the recovery. Nevertheless, it still causes some performance degradation and requires two iterations for prediction. Can we generate protein sequences in a one-shot way without loss of accuracy?

We propose PiFold (**p**rotein **i**nverse **f**olding) to address the problems above, containing a novel residue featurizer and stacked PiGNNs. As to the featurizer, for each residue, we construct more comprehensive features and introduce learnable virtual atoms to capture information overlooked by real atoms. The PiGNN considers feature dependencies at the node, edge, and global levels to learn from multi-scale residue interactions. In addition, we could completely remove the autoregressive decoder by stacking more PiGNN layers without sacrificing accuracy. Experiments show that PiFold can achieve state-of-the-art recoveries on several real-world datasets, i.e., 51.66% on CATH 4.2, 58.72 on TS50, and 60.42% on TS500. PiFold is the first graph model that could exceed 55% recovery on TS50 and 60% recovery on TS500. In addition, PiFold's inference efficiency in designing long proteins is improved by 70+ times compared to autoregressive competitors. More importantly, we conduct extensive ablation studies to reveal the role of each module, which helps deepen the reader's understanding of PiFold and may provide further inspiration for subsequent research. In summary, our contributions include:

1. We propose a novel residue featurizer to construct comprehensive residue features and learn virtual atoms to capture complementary information with real atoms.

2. We propose a PiGNN layer to learn representations from multi-scale residue interactions.

3. We suggest removing the autoregressive decoder to speed up without sacrificing accuracy.

4. We comprehensively compare advanced graph models in real-world datasets, e.g., CATH, TS50, and TS500, and demonstrate the potential of designing different protein chains.

## 2 RELATED WORKS

Recently, AI algorithms have evolved rapidly in many fields (Gao et al., 2022d; Cao et al., 2022; Tan et al., 2022a; Li et al., 2022c; He et al., 2020; Stärk et al., 2022), where the protein folding problem (Jumper et al., 2021; Wu et al., 2022; Lin et al., 2022; Mirdita et al., 2022; Wang et al., 2022; Li et al., 2022d) that has troubled humans for decades has been nearly solved. Its inverse problem- structure-based protein design - is receiving increasing attention.

**Problem definition** The structure-based protein design aims to find the amino acids sequence $\mathcal{S} = \{s_i : 1 \leq i \leq n\}$ folding into the desired structure $\mathcal{X} = \{\boldsymbol{x}_i \in \mathbb{R}^3 : 1 \leq i \leq n\}$, where $n$ is the number of residues and the natural proteins are composed by 20 types of amino acids, i.e., $1 \leq s_i \leq 20$ and $s_i \in \mathbb{N}^+$. Formally, that is to learn a function $\mathcal{F}_\theta$:

$$\mathcal{F}_\theta : \mathcal{X} \mapsto \mathcal{S}. \tag{1}$$

Because homologous proteins always share similar structures (Pearson & Sierk, 2005), the problem itself is underdetermined, i.e., the valid amino acid sequence may not be unique (Gao et al., 2020). To consider both sequential and structural dependencies, recent works (Hsu et al., 2022) suggest combining the 3D structural encoder and 1D sequence decoder, where the protein sequences are generated in an autoregressive way:

$$p(S|X;\theta) = \prod_{t=1}^{n} p(s_t|s_{<t}, X; \theta). \tag{2}$$

Based on different types of structural encoders, existing works can be divided into MLP-based, CNN-based, and GNN-based ones (Gao et al., 2022a).

**MLP-based models** MLP is used to predict the probability of 20 amino acids for each residue, and various methods are mainly difficult in feature construction. These methods are commonly evaluated on the TS50, which contains 50 native structures. For example, SPIN (Li et al., 2014) achieves 30% recovery on TS50 by using torsion angles ($\phi$ and $\psi$), sequence profiles, and energy profiles. Through adding backbone angles ($\theta$ and $\tau$), local contact number, and neighborhood distance, SPIN2 (O'Connell et al., 2018) improves the recovery to 34%. Wang's model (Wang et al., 2018) suggests using backbone dihedrals ($\phi$, $\psi$ and $\omega$), the solvent accessible surface area of backbone atoms ($C_\alpha, N, C$, and $O$), secondary structure types (helix, sheet, loop), $C_\alpha - C_\alpha$ distance and unit direction vectors of $C_\alpha - C_\alpha$, $C_\alpha - N$ and $C_\alpha - C$ and achieves 33% recovery. The MLP method enjoys a high inference speed, but suffers from a low recovery rate because the structural information is not sufficiently considered.

**CNN-based models** These methods use 2D CNN or 3d CNN to extract protein features (Torng & Altman, 2017; Boomsma & Frellsen, 2017; Weiler et al., 2018; Zhang et al., 2020a; Qi & Zhang, 2020; Chen et al., 2019) and are commonly evaluated on the TS50 and TS500. SPROF (Chen et al., 2019) adopts 2D CNN to learn residue representations from the distance matrix and achieves a 40.25% recovery on TS500. 3D CNN-based methods, such as ProDCoNN (Zhang et al., 2020a) and DenseCPD (Qi & Zhang, 2020), extract residue features from the atom distribution in a three- dimensional grid box. For each residue, after being translated and rotated to a standard position, the atomic distribution is fed to the model to learn translation- and rotation-invariant features. ProD- CoNN (Zhang et al., 2020a) designs a nine-layer 3D CNN with multi-scale convolution kernels and achieves 42.2% recovery on TS500. DenseCPD (Qi & Zhang, 2020) uses the DensetNet architec- ture (Huang et al., 2017) to boost the recovery to 55.53% on TS500. Recent works (Anand et al., 2022) have also explored the potential of deep models to generalize to *de novo* proteins. Despite the improved recovery achieved by the 3D CNN models, their inference is slow, probably because they require separate preprocessing and prediction for each residue.

**Graph-based models** These methods use $k$-NN graph to represent the 3D structure and employ graph neural networks (Defferrard et al., 2016; Kipf & Welling, 2016; Veličković et al., 2017; Zhou et al., 2020; Zhang et al., 2020b; Gao et al., 2022b; Tan et al., 2022b; Gao et al., 2022c) to extract residue features while considering structural constraints. The protein graph encodes residue infor- mation and pairwise interactions as the node and edge features, respectively. GraphTrans (Ingraham et al., 2019) uses the graph attention encoder and autoregressive decoder for protein design. GVP (Jing et al., 2020) proposes geometric vector perceptrons to learn from both scalar and vector fea- tures. GCA (Tan et al., 2022c) introduces global graph attention for learning contextual features. In

addition, ProteinSolver (Strokach et al., 2020) is developed for scenarios where partial sequences are known while does not report results on standard benchmarks. Recently, AlphaDesign (Gao et al., 2022a), ProteinMPNN (Dauparas et al., 2022) and Inverse Folding (Hsu et al., 2022) achieve dramatic improvements. Compared to CNN methods, graph models do not require rotating each residue separately as in CNN, thus improving the training efficiency. Compared to MLP methods, the well-exploited structural information helps GNN obtain higher recovery. In summary, GNN can achieve a good balance between efficiency and accuracy.

## 3 METHOD

### 3.1 OVERALL FRAMEWORK

We show the overall PiFold framework in Figure.2, where the inputs are protein structures, and outputs are protein sequences expected to fold into the input structures. We propose a novel residue featurizer and PiGNN layer to learn expressive residue representations. Specifically, the residue featurizer constructs comprehensive residue features and creates learnable virtual atoms to capture information complementary to real atoms. The PiGNN considers multi-scale residue interactions in node, edge, and global context levels. PiFold could generate protein sequences in a one-shot manner with a higher recovery than previous autoregressive (Ingraham et al., 2019; Jing et al., 2020; Tan et al., 2022c; Dauparas et al., 2022; Hsu et al., 2022) or iterative models (Gao et al., 2022a).

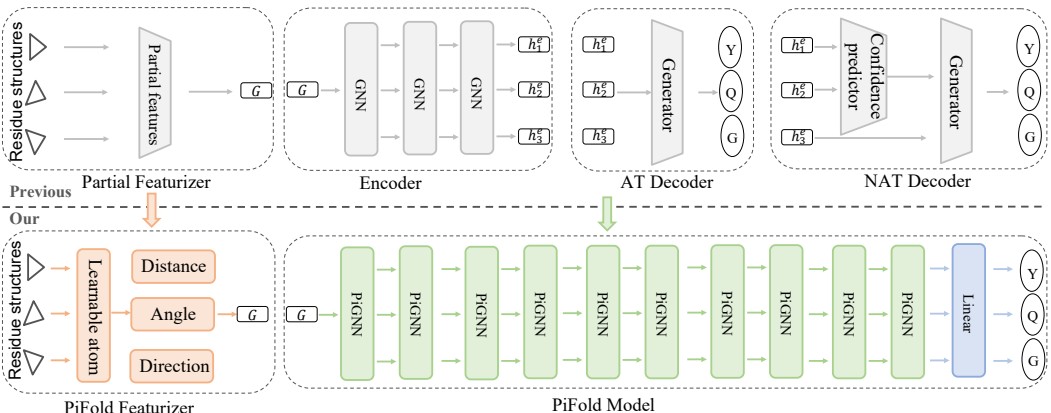

**Figure 2:** The overall framework of PiFold. The featurizer module extracts comprehensive residue features from the fixed real atoms and learnable virtual atoms. We stack ten layers of PiGNNs, considering node-level, edge-level, and global-level residue interactions, to learn expressive residue representations. PiFold generates protein sequences in a one-shot manner, without autoregressive or iterative decoding schemas.

### 3.2 FEATURIZE MODULE

This section describes how to construct protein graphs and residue features. We recommend readers view Fig.3 for intuitive understanding.

**Graph structure** We represent the protein as a $k$-NN graph derived from residues to consider the 3D dependencies, where $k$ defaults to 30. The protein graph $\mathcal{G}(A, X, E)$ consists of the adjacency matrix $A \in \{0,1\}^{n,n}$, node features $X \in \mathbb{R}^{n,f_n}$, and edge features $E \in \mathbb{R}^{m,f_e}$. Note that $n$ and $m$ are the numbers of nodes and edges, and we construct $f_n$ node features and $f_e$ edge features considering the inductive bias of residues' stable structure, order, and coordinates. Since each residue consists of $C_\alpha, C, N$ and $O$, we could construct local coordinate system $Q_i = [\boldsymbol{b}_i, \boldsymbol{n}_i, \boldsymbol{b}_i \times \boldsymbol{n}_i]$ for residue $i$, where $\boldsymbol{u}_i = C_{\alpha_i} - N_i, \boldsymbol{v}_i = C_i - C_{\alpha_i}, \boldsymbol{b}_i = \frac{\boldsymbol{u}_i - \boldsymbol{v}_i}{||\boldsymbol{u}_i - \boldsymbol{v}_i||}$, and $\boldsymbol{n}_i = \frac{\boldsymbol{u}_i \times \boldsymbol{v}_i}{||\boldsymbol{u}_i \times \boldsymbol{v}_i||}$. Based on these coordinate systems, we could construct rotation- and translation-invariant features for single or pairs of residues, including distance, angle, and direction; refer to Fig.3.

**Distance features** Given atom pairs $A$ and $B$, the distance feature is RBF($||A - B||$), where RBF is a radial basis function. Note that $A \in \{C_i, C_{\alpha_i}, N_i, O_i\}, B \in \{C_i, C_{\alpha_i}, N_i, O_i\}$ when computing features for residue $i$, while $A \in \{C_i, C_{\alpha_i}, N_i, O_i\}$ and $B \in \{C_j, C_{\alpha_j}, N_j, O_j\}$ when computing

features for residue pair $(i, j)$. Careful readers should notice that the number of features is proportional to the square of the number of atoms, which will be further analyzed and simplified in Table.6. Beyond the distances between real atoms, we also introduce virtual atoms $\{V_i^1, V_i^2, V_i^3, \cdots\}$ for residue $i$. The additional node and edge features are generated by sampling $A$ and $B$ from $\{V_i^1, V_i^2, V_i^3, \cdots\} \times \{V_i^1, V_i^2, V_i^3, \cdots\}$ and $\{V_i^1, V_i^2, V_i^3, \cdots\} \times \{V_j^1, V_j^2, V_j^3, \cdots\}$, respectively. To ensure that the virtual atoms are translation- and rotation- equivariant to the protein structure, the model is required to learn their relative position in each local coordinate system. The relative position vectors are shared across all residues, thus forcing the model to learn consensus knowledge to extend to general scenarios. We normalize the relative coordinates by limiting their length to a unit vector for stable training. Formally, denoting the relative position of the $k$-th virtual atom is $(x_k, y_k, z_k)$, we compute the absolute coordinates of $V_i^k$ by

$$\begin{cases} V_i^k = [x_k \boldsymbol{b}_i, y_k \boldsymbol{n}_i, z_k(\boldsymbol{b}_i \times \boldsymbol{n}_i)] + C_{\alpha_i} \\ x_k^2 + y_k^2 + z_k^2 = 1, \end{cases} \tag{3}$$

where $x_k, y_k, z_k$ are learnable parameters.

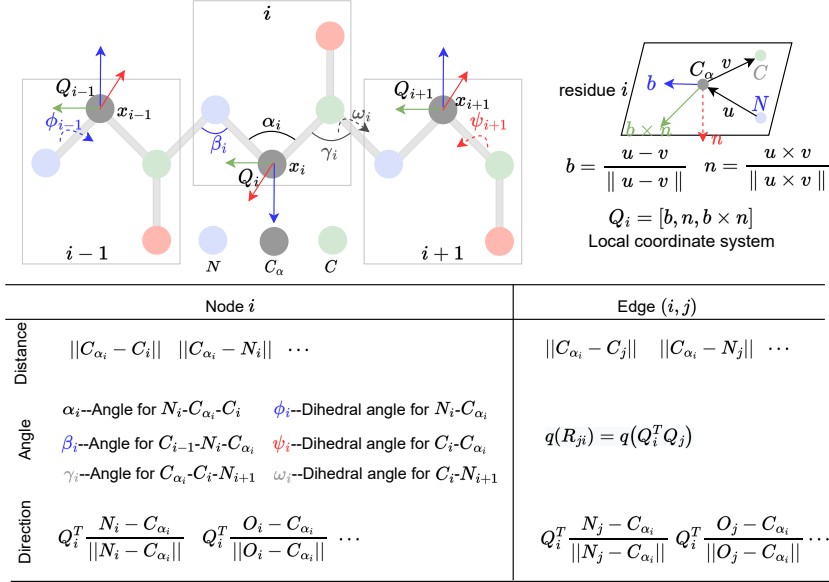

**Figure 3:** PiFold featurizer. We construct distance, angle, and direction features for single or paired residues, all invariant to rotation and translation. Learnable virtual atoms are used for additional distance features.

**Angle features**  As shown in Figure.3, we use the bond angles $(\alpha_i, \beta_i, \gamma_i)$ and torsion angles $(\phi_i, \psi_i, \omega_i)$ as node features considering the backbone order and local geometry between adjacent residues. For residue pairs $(i, j)$, the angular features represent the quaternions of relative rotation between their local coordinate systems, written as $q_{ij} = q(Q_i^T Q_j)$, where $q_{ij}$ is the quaternion encoding function.

**Direction features**  For single residue $i$, we use the directions of inner atoms to $C_{\alpha_i}$ as node features. For example, the direction feature of $N_i$ is $Q_i^T \frac{N_i - C_{\alpha_i}}{||N_i - C_{\alpha_i}||}$. Similarly, for pairs of residues $(i, j)$, the directions include all atoms of residue $j$ to the $C_{\alpha_i}$, and the direction of $N_j$ is $Q_i^T \frac{N_j - C_{\alpha_i}}{||N_j - C_{\alpha_i}||}$. Note that we use the relative direction concerning the local coordinate system $Q_i$ to make the features rotationally invariant.

### 3.3  PiGNN

We propose the PiGNN layer to learn geometric residue representations considering multi-scale residue interactions in node, edge, and global context levels.

**Local node interactions**  We use the simplified graph transformer (Gao et al., 2022a) to update node representations, where the model learns multi-headed attention weights via a MLP instead of

separately computing $Q$ and $K$. Formally, we take $\boldsymbol{h}_i^l$ and $\boldsymbol{e}_{ji}^l$ as the output feature vectors of node $i$ and edge $j \to i$ in layer $l$. Before the first GNN layer, we use MLPs to project all input features to the $d$-dimensional space, thus $\boldsymbol{h}_i^0 \in \mathbb{R}^d$ and $\boldsymbol{e}_{ji}^0 \in \mathbb{R}^d$. When considering the attention mechanisms centered in node $i$, the attention weight $a_{ji}$ at the $l+1$ layer is calculated by:

$$\begin{cases} w_{ji} = \text{AttMLP}(\boldsymbol{h}_j^l || \boldsymbol{e}_{ji}^l || \boldsymbol{h}_i^l) \\ a_{ji} = \frac{\exp w_{ji}}{\sum_{k \in \mathcal{N}_i} \exp w_{ki}} \end{cases} \tag{4}$$

where $\mathcal{N}_i$ is the neighborhood system of node $i$ and $||$ means the concatenation operation. The node feature $\boldsymbol{h}_i^l$ is updated by:

$$\begin{cases} \boldsymbol{v}_j = \text{NodeMLP}(\boldsymbol{e}_{ji}^l || \boldsymbol{h}_j^l) \\ \hat{\boldsymbol{h}}_i^{l+1} = \sum_{j \in \mathcal{N}_i} a_{ji} \boldsymbol{v}_j \end{cases} \tag{5}$$

**Local edge interactions** The protein graph is an attributed graph containing both node and edge features; however, the simplified graph transformer is a node-centered network, which does not iteratively update edge features. We find that this neglect will lead to suboptimal representations, and introducing the edge updating layer could improve the model capability:

$$\boldsymbol{e}_{ji}^{l+1} = \text{EdgeMLP}(\hat{\boldsymbol{h}}_j^l || \boldsymbol{e}_{ji}^l || \hat{\boldsymbol{h}}_i^l) \tag{6}$$

**Global context attention** While the local interactions play a crucial role in learning residue representations (Ingraham et al., 2019; Jing et al., 2020; Gao et al., 2022a), the global information (Tan et al., 2022c) is also proved to be valuable for improving protein design. However, the time complexity of global attention across the whole protein is proportional to the square of the protein length, which significantly increases the computational overhead. To simultaneously enjoy the improved recovery and with good efficiency, we suggest learning a global context vector for each protein and using it to apply gated attention for node representations:

$$\begin{cases} \boldsymbol{c}_i = \text{Mean}(\{\hat{\boldsymbol{h}}_k^{l+1}\}_{k \in \mathcal{B}_i}) \\ \boldsymbol{h}_i^{l+1} = \hat{\boldsymbol{h}}_i^{l+1} \odot \sigma(\text{GateMLP}(\boldsymbol{c}_i)) \end{cases} \tag{7}$$

where $\mathcal{B}_i$ is the index set of residues belonging to the same protein as residue $i$, $\odot$ is element-wise product operation, and $\sigma(\cdot)$ is the sigmoid function. The computational cost of the global context attention is linear to the residue numbers.

## 4 EXPERIMENTS

We comprehensively evaluate PiFold on different datasets and show its superior accuracy and efficiency. Detailed ablation studies are applied to deepen the understanding of PiFold. In summary, we will answer the following questions:

- **Performance (Q1):** Could PiFold achieve SOTA accuracy and efficiency on CATH dataset?
- **Ablation (Q2):** How much gain can be obtained for each improvement?
- **Benchmarking (Q3):** Could PiFold be extended to more benchmarks, such as TS50 and TS500?

### 4.1 PERFORMANCE (Q1)

**Objective & Setting** Could PiFold achieve SOTA recovery and efficiency on real-world datasets? Does the efficiency improvement require sacrificing the generative quality? To answer these questions, we compare PiFold against recent strong baselines on the CATH (Orengo et al., 1997) dataset. We use the same data splitting as GraphTrans (Ingraham et al., 2019) and GVP (Jing et al., 2020), where proteins are partitioned by the CATH topology classification, resulting in 18024 proteins for training, 608 proteins for validation, and 1120 proteins for testing. We stack ten layers of PiGNN to construct the PiFold model with hidden dimension 128. The model is trained up to 100 epochs by default using the Adam optimizer on NVIDIA V100s. The batch size and learning rate used for training are 8 and 0.001, respectively. For studying the generative quality, we report perplexity and median recovery scores on short-chain, single-chain, and all-chain settings. Regarding efficiency, we report the model inference time when generating 100 long chains with an average length of 1632.

**Baselines** We compare PiFold with recent graph models, including StructGNN, StructTrans (Ingraham et al., 2019), GCA (Tan et al., 2022c), GVP (Jing et al., 2020), GVP-large, AlphaDesign (Gao et al., 2022a), ESM-IF (Hsu et al., 2022), ProteinMPNN (Dauparas et al., 2022), since most of them are open-source. To make a fair and reliable comparison, we reproduce StructGNN, Struct-Trans, GCA, GVP, AlphaDesign, and ProteinMPNN under the same data splitting on CATH 4.2. The relevant results were generally consistent with their papers. Because GVP-large and ESM-IF do not provide training scripts and their code framework is different from other methods, we have difficulty reproducing them and therefore copy the results from their manuscripts. It should be reminded that ESM-IF uses CATH 4.3 for training, which may lead to incomparable recovery and perplexity due to the different data volumes. MLP and CNN-based methods are ignored here because most of them do not report results on CATH nor provide publicly available codes, and we will discuss them in Section.4.3. For more baseline details, we suggest the reader refer to the Appendix.

**Table 1:** Results comparison on the CATH dataset. All baselines are reproduced under the same code framework, except ones marked with †. We copy results of GVP-large and ESM-IF from their manuscripts (Hsu et al., 2022). The **best** and suboptimal results are labeled with bold and underline.

| Model | Perplexity ↓ | | | Recovery % ↑ | | | CATH version | |
| | Short | Single-chain | All | Short | Single-chain | All | 4.2 | 4.3 |
| --- | --- | --- | --- | --- | --- | --- | --- | --- |
| StructGNN | 8.29 | 8.74 | 6.40 | 29.44 | 28.26 | 35.91 | ✓ | |
| GraphTrans | 8.39 | 8.83 | 6.63 | 28.14 | 28.46 | 35.82 | ✓ | |
| GCA | 7.09 | 7.49 | 6.05 | 32.62 | 31.10 | 37.64 | ✓ | |
| GVP | 7.23 | 7.84 | 5.36 | 30.60 | 28.95 | 39.47 | ✓ | |
| GVP-large† | 7.68 | **6.12** | 6.17 | 32.6 | **39.4** | 39.2 | | ✓ |
| AlphaDesign | 7.32 | 7.63 | 6.30 | 34.16 | 32.66 | 41.31 | ✓ | |
| ESM-IF† | 8.18 | 6.33 | 6.44 | 31.3 | 38.5 | 38.3 | | ✓ |
| ProteinMPNN | 6.21 | 6.68 | 4.61 | 36.35 | 34.43 | 45.96 | ✓ | |
| PiFold | **6.04** | 6.31 | **4.55** | **39.84** | 38.53 | **51.66** | ✓ | |

**Recovery** We report the perplexity and recovery score in Table.1, where two subsets of the full test set are also considered, following previous works (Ingraham et al., 2019; Jing et al., 2020; Tan et al., 2022c; Hsu et al., 2022). The "Short" set contains proteins up to length 100, and the "Single chain" contains proteins recorded as single chains in the Protein Data Bank. We observe that the proposed PiFold could consistently improve the perplexity (lower is better) and recovery score (higher is better) across different test sets. On the "Short" and "All" datasets, PiFold achieves the best perplexities and recovery scores, where the recovery improvement are 3.49% and 1.94%, respectively. On the "Single-chain" dataset, PiFold is the best model when trained on CATH 4.2.

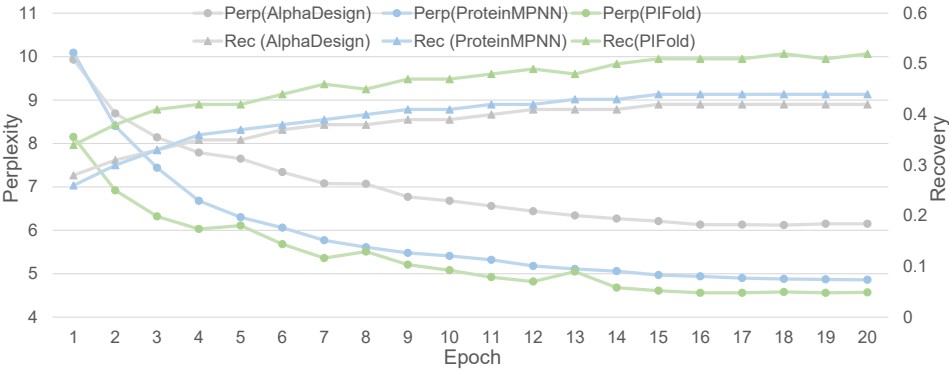

**Figure 4:** Training PiFold for 20 epochs is enough to achieve SOTA performance.

**Efficiency** We already know that PiFold can achieve higher recovery; does this improvement come at the expense of efficiency? We further assess the training and inference time costs of PiFold and the competitive baselines (AlphaDesign and ProteinMPNN). PiFold could achieve state-of-the-art perplexity and recovery during the training phase with fewer epochs. As shown in Figure.4, 20 epochs are enough for PiFold. As to the inference speed, all baselines except AlphaDesign adopt an autoregressive decoder to generate residues one by one, and the computational complexity is O(L)

to the protein length L. However, PiFold enjoys O(1) computational complexity due to the one-shot generative schema. We benchmark the inference speed of different graph models on 100 long chains with an average length of 1632 and show the results in Fig.1, where PiFold is 70 times faster than autoregressive competitors, including ProteinMPNN, ESM-IF, GCA, StructGNN, StructTrans, GVP.

## 4.2 ABLATION (Q2)

**Objective & Setting**   How much gain could be obtained from the introduced features and modules? Will removing the autoregressive decoder reduce performance? We conduct comprehensive ablation studies for the features and modules to answer these questions. All models are trained up to 20 epochs with a learning rate of 0.001 using the Adam optimizer and OneCycle scheduler.

**Table 2:** Ablation studies of feature types. The lower the ranking number, the higher the recovery rate and the model performs better.

| | Model | PiFold | model 1 | model 2 | model 3 | model 4 | model 5 | model 6 |
|---|---|---|---|---|---|---|---|---|
| | Distance | ✓ | | ✓ | ✓ | ✓ | ✓ | ✓ |
| Node | Angle | ✓ | ✓ | | ✓ | ✓ | ✓ | ✓ |
| | Direction | ✓ | ✓ | ✓ | | ✓ | ✓ | ✓ |
| | Distance | ✓ | ✓ | ✓ | ✓ | | ✓ | ✓ |
| Edge | Angle | ✓ | ✓ | ✓ | ✓ | ✓ | | ✓ |
| | Direction | ✓ | ✓ | ✓ | ✓ | ✓ | ✓ | |
| Results | Perplexity ↓ | 4.55 | 4.59 | 4.62 | 4.53 | **4.80** | 4.54 | 4.65 |
| | Recovery ↑ | 51.66 | 51.49 | 51.05 | 51.61 | **49.61** | 51.51 | 50.86 |
| Summary | Rank | – | 5 | 3 | 6 | **1** | 4 | 2 |

**Feature type ablation**   We determine the contribution rank of different types of features in Table.2. We find that edge features are more important than node features, suggesting that amino acid types may be inferred primarily from residue interactions. Another discovery is that edge distances are the essential features; however, the number of distances is proportional to the square of the atom numbers, which may lead to feature explosion and redundancy. We further analyze the role of each edge distance in Table.6, where the "$C_\alpha$-O" and "C-N" distances are the most significant ones. In addition, the ablation study in Table.6 also demonstrates that virtual atoms can lead to performance gains, and the more virtual atoms, the higher recovery. One possible explanation is that virtual atoms can learn valuable information complementary to backbone atoms, e.g., side chain direction.

**Table 3:** Ablation of PiGNN modules.

| | | PiFold | model 1 | model 2 | model 3 | model 4 | model 5 |
|---|---|---|---|---|---|---|---|
| | GCN | | ✓ | | | | |
| Node | GAT | | | ✓ | | | |
| | QKV | | | | ✓ | | |
| | AttMLP | ✓ | | | | ✓ | ✓ |
| Edge | UpdateEdge | ✓ | ✓ | ✓ | ✓ | ✓ | |
| Global | ContextAtt | ✓ | ✓ | ✓ | ✓ | | |
| Results | Perplexity | 4.55 | 4.74 | 4.96 | 4.54 | 4.62 | 4.76 |
| | Recovery | 51.66 | 49.65 | 45.78 | 50.74 | 51.22 | 50.00 |
| Summary | Change | – | ↓↓ | ↓↓↓ | ↓↓ | ↓ | ↓↓ |

**Model ablation**   We conduct systematic experiments to determine whether modules of PiGNN are effective and reveal their relative importance. At to the node message passing module, we compare the node message passing layer (AttMLP) against GCN (Kipf & Welling, 2016), GAT (Veličković et al., 2017) and QKV-based attention layer (Ingraham et al., 2019; Tan et al., 2022c; Dauparas et al., 2022). We also investigate whether the edge updating operation and global context attention could contribute to the model performance. As shown in Table.3, we find that the AttMLP-based messaging mechanism can improve the recovery by at least 0.9%. In addition, the global context attention and edge updating could improve the recovery by 0.44% and 1.22%, respectively.

**AT ablation**   Could PiFold remove the autoregressive mechanism and still ensure good performance? We further explore the effectiveness of the autoregressive decoder (Ingraham et al., 2019) and one-shot PiGNN encoder. As shown in Table.4, by gradually replacing the autoregressive decoder with the one-shot PiGNN (model 1 → model 5), the recovery will improve. At the same time, the inference time cost on CATH will reduce. This phenomenon suggests that the expressive encoder is more important than the autoregressive decoder. The recovery score can be further improved by using a deeper encoder (model 5 → model 6 → PiFold).

**Table 4:** Autoregressive ablation. "Enc" and "AT" indicate the encoder's and autoregressive layers' numbers.

|  | PiFold | model 1 | model 2 | model 3 | model 4 | model 5 | model 6 |
|---|---|---|---|---|---|---|---|
| Enc | 10 | 3 | 3 | 4 | 5 | 6 | 8 |
| AT | 0 | 2 | 3 | 2 | 1 | 0 | 0 |
| Perplexity ↓ | 4.55 | 5.06 | 5.04 | 4.92 | 4.83 | 4.58 | 4.53 |
| Recovery ↑ | 51.66 | 49.30 | 49.37 | 49.60 | 50.41 | 50.96 | 51.22 |
| Test Time ↓ | 36s | 527s | 707s | 522s | 347s | 30s | 34s |

## 4.3 GENERALIZATION (Q3)

**Table 5:** Results on TS50 and TS500. All baselines are reproduced under the same code framework, except ones marked with †, whose results are copied from their manuscripts. The **best** and suboptimal results are labeled with bold and underline.

| Group | Model | TS50 | | | TS500 | | |
|---|---|---|---|---|---|---|---|
| | | Perplexity ↓ | Recovery ↑ | Worst ↑ | Perplexity ↓ | Recovery ↑ | Worst ↑ |
| MLP | SPIN † | | 30.30 | | | 30.30 | |
| | SPIN2 † | | 33.60 | | | 36.60 | |
| | Wang's model † | | 33.00 | | | 36.14 | |
| CNN | SPROF † | | 39.16 | | | 40.25 | |
| | ProDCoNN † | | 40.69 | | | 42.20 | |
| | DenseCPD † | | 50.71 | | | 55.53 | |
| Graph | StructGNN | 5.40 | 43.89 | 26.92 | 4.98 | 45.69 | 0.05 |
| | GraphTrans | 5.60 | 42.20 | 29.22 | 5.16 | 44.66 | 0.03 |
| | GVP | 4.71 | 44.14 | 33.73 | 4.20 | 49.14 | **0.09** |
| | GCA | 5.09 | 47.02 | 28.87 | 4.72 | 47.74 | 0.03 |
| | AlphaDesign | 5.25 | 48.36 | 32.31 | 4.93 | 49.23 | 0.03 |
| | ProteinMPNN | 3.93 | 54.43 | 37.24 | 3.53 | 58.08 | 0.03 |
| | PiFold (our) | **3.86** | **58.72** | **37.93** | **3.44** | **60.42** | 0.03 |

**Objective & Results** To present a comprehensive comparison and assess the generalizability of PiFold across different data sets, we report results on TS50 and TS500 in Table.5. We reproduce graph-based models and show the perplexity, median recovery score, and worst recovery. For MLP and CNN models, the results are copied from their manuscripts. We summarize that PiFold could achieve consistent improvements across TS50 and TS500. To our knowledge, PiFold is the first graph model that could exceed 55% recovery on TS50 and 60% on TS500. Finally, in Figure. 5, we show the potential of PiFold in designing all-alpha, all-beta, and mixed proteins.

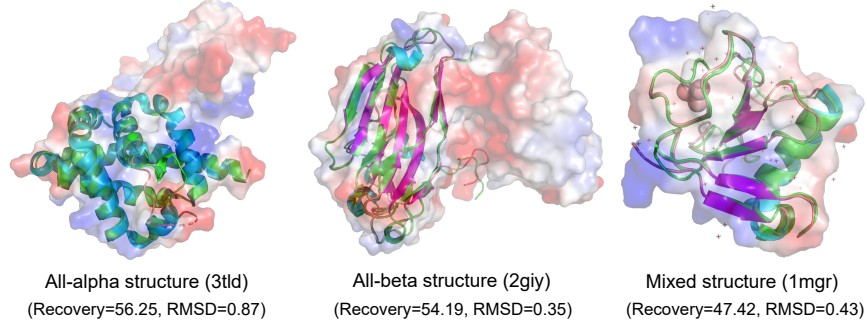

| All-alpha structure (3tld) | All-beta structure (2giy) | Mixed structure (1mgr) |
|---|---|---|
| (Recovery=56.25, RMSD=0.87) | (Recovery=54.19, RMSD=0.35) | (Recovery=47.42, RMSD=0.43) |

**Figure 5:** Visual examples. For native structures, we color Helix, Sheet, and Loop with cyan, magenta, and orange, respectively. We use AlphaFold2 to predict the structures of designed sequences, marked in green. We provide the recovery score and structural RMSD relative to the ground truth proteins.

## 5 CONCLUSION

PiFold, as an AI design method for structure-based protein design, significantly improves the recovery and greatly increases the efficiency simultaneously, by using the proposed protein featurizer and PiGNN. It achieves 51.66%, 58.72%, and 60.42% recovery scores on CATH 4.2, TS50, and TS500 test sets, respectively, and is 70 times faster in inference speed than autoregressive competitors for designing long proteins. We also conduct comprehensive ablation to inspire future studies.

## 6    ACKNOWLEDGEMENTS

We thank the anonymous reviewers for their constructive and helpful reviews. This work was supported by the National Key R&D Program of China (Project 2022ZD0115100), the National Natural Science Foundation of China (Project U21A20427), the Research Center for Industries of the Future (Project WU2022C043), and the Westlake University Research Funds from the Center for Synthetic Biology and Integrated Bioengineering (Project WU2022A009).

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

# A  APPENDIX

**Distance ablation**  We further analyze the role of each edge distance in Table.6, where the $C_\alpha$-O"
and "C-N" distances are the most significant ones. Last but not least, we find that the distances
between learnable virtual atoms significantly improve the recovery, suggesting that the model can
automatically discover critical residual points.

**Table 6:** Ablation of edge distance features

| | Model | PiFold | 1 | 2 | 3 | 4 | 5 | 6 | 7 | 8 | 9 | 10 |
|---|---|---|---|---|---|---|---|---|---|---|---|---|
| Real distances | $C_\alpha$-$C_\alpha$ | ✓ | ✓ | | | | | | | ✓ | ✓ | ✓ |
| | $C_\alpha$-C | ✓ | | ✓ | ✓ | ✓ | | | | ✓ | ✓ | ✓ |
| | $C_\alpha$-N | ✓ | | ✓ | | | | | | ✓ | ✓ | ✓ |
| | $C_\alpha$-O | ✓ | | ✓ | ✓ | | ✓ | | | ✓ | ✓ | ✓ |
| | C-C | ✓ | ✓ | | | | | | | ✓ | ✓ | ✓ |
| | C-N | ✓ | | ✓ | ✓ | | | ✓ | | ✓ | ✓ | ✓ |
| | C-O | ✓ | | ✓ | | | | | | ✓ | ✓ | ✓ |
| | N-N | ✓ | ✓ | | | | | | | ✓ | ✓ | ✓ |
| | N-O | ✓ | | ✓ | ✓ | | | | ✓ | ✓ | ✓ | ✓ |
| | O-O | ✓ | ✓ | | | | | | | ✓ | ✓ | ✓ |
| Virtual atoms | 0 | | | | | | | | | ✓ | | |
| | 1 | | | | | | | | | | ✓ | |
| | 2 | | | | | | | | | | | ✓ |
| | 3 | ✓ | ✓ | ✓ | ✓ | ✓ | ✓ | ✓ | ✓ | | | |
| Results | Perplexity ↓ | 4.55 | 4.55 | 4.53 | 4.53 | 4.54 | 4.54 | 4.52 | 4.55 | 4.59 | 4.54 | 4.53 |
| | Recovery ↑ | 51.66 | 51.52 | 51.63 | 51.74 | 51.49 | **51.70** | **51.89** | 51.50 | 51.02 | 51.10 | 51.51 |
| | Change | − | ↓ | ↓ | ↑ | ↓ | ↑ | ↑ | ↓ | ↓↓↓ | ↓↓ | ↓ |

**Baseline details**  ProteinMPNN supports the design of proteins when partial residues are known,
while we focus on designing sequences from structures without any known residues. On CATH4.2,
we note that ProteinMPNN directly assigns sequence labels to residues with missing coordinates,
indicating these residues are treated as known ones. If we allow this trick, ProteinMPNN could
achieve 51% recovery on CATH4.2. To force all methods to be compared under the same setting,
we remove this operation and update the recovery to 45.96%. For ESM-IF, we report their results
without training by third-party data. They have proved that increased data volume could significantly
improve model capability, thus affecting researchers' judgments about the models. We reproduced
most of the baselines and allow them to be trained and evaluated under the same data splitting and
code framework. We admire these works, and thank all the authors providing open-source codes.

