# OpenReview forum: "PiFold: Toward effective and efficient protein inverse folding"
_ICLR.cc/2023/Conference — ICLR 2023 notable top 25%_

### Official Review · Reviewer_DZpe · 2022-10-24

**Confidence:** 4
**Correctness:** 4
**Technical Novelty And Significance:** 3
**Empirical Novelty And Significance:** 3
**Recommendation:** 8

**Clarity, Quality, Novelty And Reproducibility:**

The paper is written in a clear way. However, some of the figures lack cohesion, and some terms are used without explanation:
- Figure 2 PiGNN is written "ProGNN"?
- Sec 4 CATH misspelled CTAH
- Table 2: Rank is confusing, maybe explain that the rank corresponds to the highest drop in performace in the caption
- Figure 2: What are the AT Decoder and NAT Decoders?

**Strength And Weaknesses:**

Strengths:
- PiFold is compared to a suite of recent protein design architectures
- The training and testing splits are solid and makes comparison to previous works easy.
- The model shows great sequence recovery performance and speed improvements.
- The paper is written in a concise manner and understanding it is easy

Weaknesses:
- The ablations are interesting but slightly lacking - the "leave one out" approach seems imply no one change makes a big difference, PiFold is still state of the art despite ablating so many different changes. Maybe a better ablation is to show which change is making the difference here, and giving it such a large gap between PiFold and other competitors.

**Summary Of The Paper:**

- PiFold is a system for fixed backbone sequence design with a new PiGNN layer that combine node and edge information.
- By using a parallel decoder, the model is able to make predictions many times faster
- Using more features in the encoder, PiFold is able to still obtain state of the art results.

**Summary Of The Review:**

The paper is technically strong, the writing could improve slightly, but overall I would recommend an accept.

---

### Official Review · Reviewer_4ek8 · 2022-10-24

**Confidence:** 3
**Correctness:** 3
**Technical Novelty And Significance:** 2
**Empirical Novelty And Significance:** 3
**Recommendation:** 6

**Clarity, Quality, Novelty And Reproducibility:**

Clarity: The paper is clear in most parts. More background on protein folding might be necessary for a general ML audience and would be a good addition to the paper to improve readability.
Quality: The paper's core contribution appears to be the empirical performance of the proposed algorithm, which is well supported by the experiments and the ablation studies (if the experiments are reproducible).
Novelty: The novelty of the method is somewhat limited as most components can be found in previous work.
Reproducibility: no repository or zipped code has been provided.

Minor comments:

- Some parts of Related Works matches the AlphaDesign paper's Related work section word-to-word and should be rephrased.

- Typo in Related Works: "sovlent-accessible"

- Typo in Table 3: "UpadateEdge"

**Strength And Weaknesses:**

Strengths:

- The paper is well-written and organized in a logical manner. It is easy to follow in most parts.

- The algorithm shows good performance on the benchmarks. It consistently outperforms other methods in terms of recovery and perplexity in most cases and has the fastest inference time.

- The ablation studies are in-depth, authors put great effort into verifying their design choices.


Weaknesses:

- The source code is missing, and therefore the results are not reproducible.

- The novelty compared to AlphaDesign and GCA is somewhat limited. One contribution the authors claim is the way features are generated. Here the novelty appears to be the introduction of virtual atoms, other elements can be found in AlphaDesign. The second contribution claimed by the authors is the PiGNN module, however this is based on SGT (in case of modeling local interactions) which has been introduced in AlphaDesign as well. Extracting global context has also been done in GCA in a similar manner. Could authors clarify the original contribution compared with AlphaDesign and GCA?

- Questions/concerns with respect to the experiments:

  - How similar are CATH 4.2 and 4.3? ESM-IF has only been evaluated on 4.3 and I am unsure how meaningful it is to include those results in Table 1 as a basis of comparison with the proposed method given they are evaluated on different datasets.

  - Table 4 shows that when the number of encoders increases from 6 to 8, test time increases by 4 seconds and recovery increases by 0.26%. However, when the number of encoders is increased from 8 to 10 (PiFold), test time only increases by 2 seconds. Furthermore, recovery increases by 0.44% which is almost doubled compared with the increase from 6 to 8 encoders. Can authors explain what causes the significant difference from linear scaling in test time? How does test time scale with the number of encoders? Furthermore, I would expect either linear or saturating behavior of recovery with respect to number of encoders, but this is not what we see in Table 4. Could authors discuss the performance scaling with # of encoders? Why exactly 10 encoders have been used in the final architecture?

**Summary Of The Paper:**

This paper proposes a method for structure-based protein design that utilizes graph neural networks to capture local and global interactions between atoms in the protein and outputs a sequence of amino acids with the desired input structure. Authors introduce a method to generate useful features from the input structure. These features derived from real atoms and learned virtual atoms are then processed through a stack of attention and MLP-based blocks that can directly generate the amino acid sequence. Through numerical experiments on benchmark datasets, authors demonstrate the performance and efficacy of their algorithm, yielding state-of-the-art recovery with low inference time.

**Summary Of The Review:**

Overall, I am leaning towards borderline rejecting the paper, as the results are not reproducible and the original contribution compared to some previous work is somewhat unclear (see both under Weaknesses).

---

### Official Review · Reviewer_cTW6 · 2022-10-31

**Confidence:** 4
**Clarity, Quality, Novelty And Reproducibility:** n/a
**Correctness:** 3
**Technical Novelty And Significance:** 3
**Empirical Novelty And Significance:** 3
**Recommendation:** 6

**Strength And Weaknesses:**

Overall, this paper is well-motivated and easy to follow. Extensive experiments are also presented.

However, my major concerns come from the experiments and comparisons, which are performed less comprehensively and are not convincing. Please refer to the questions.

**Questions:**
1. The respective contributions of the proposed featurizer and PiGNN Layer remain unclear. How would it be if using the proposed features upon other strong models, e.g., protein mpnn? Besides, can PiGNN layers also help improves GVP GNN if placing them upon GVP features and encoders?
2. Ablation study wrt autoregressivity (Tab 4) seems somewhat unfair, where PiFold uses 10 layers while all the AT competitors use much fewer layers (#Enc + #AT in total, the maximum number of layers are 6 < 10). What’s more, model 4 consisting of 4 Enc and 1 AT decoder performs very closely to PiFold (~-1%). For more convincing conclusions, authors should conduct experiments with AT-based models that must integrate the same number of layers as PiFold, and ablate the proportion btw Enc and AT.

It would be great if these results would be included during the author response period.

Besides, I also have some other questions:

1. How about the scale of parameters of the proposed model compared to the previous ones? Plus, what would it be if scaling up the size of the parameters? How about scaling up data size (e.g., using alphafold-predicted data like in ESM-IF)?
2. In most NLP studies, non-autoregressive models are known to lag behind their autoregressive counterparts considerably. However in this study, the proposed non-autoregressive models, though the comparisons are not that fair as I just pointed out, seem to do a really good job and even beat the autoregressive ones. What are the reasons for such differences in these two areas?
3. Could you further provide results on CATH 4.3 so as to compare with the SOTA method ESM-IF (and their variant with AF2 data augmentation)

**Minors:**

- Sec. 1, page 2, “Experiments show that Pi- Fold can achieve state-of-the-art recoveries on several real-world datasets, i.g., 51.66% on CATH 4.2, 58.72 on TS 50, and 60.42% on TS 500.”: “i.g.” → “e.g.”
- Symbols of quotes are misused across the manuscript. For instance, see paragraph Recovery in page 7, and many others.

**Summary Of The Paper:**

This paper studies the protein inverse folding problem, i.e., predicting the most probable amino acid sequence given protein structure. To this end, they propose to improve two aspects of this task, one is more informative structural features for better representing protein structures, and the other one is a one-shot sequence decoder, PiGNN, for a more efficient generation. Experiments show that the proposed approach, the PiFold, achieves faster and more accurate performance for inverse folding.



**Summary Of The Review:**

Overall, I think this paper is mostly well presented, and the contributions can be considered a bit novel. But the empirical evaluations remain poor and not sufficiently convincing. As a  result, I lean to consider this as a boardline work. Acceptance would be possible upon author responses and more comprehensive and fair experimental evidence.

---

### Decision · Program_Chairs · 2023-01-20

**Decision:**

Accept: notable-top-25%

**Justification For Why Not Higher Score:**

The paper makes some exciting contributions and provide a great evaluation of the proposed approach. However, the contributions are not groundbreaking (e.g. the approach still follows the usual GNN-based paradigm for inverse protein folding) to warrant oral presentation.

**Justification For Why Not Lower Score:**

This paper makes novel methodological contributions on protein inverse folding (e.g. the introduction of virtual atoms etc) and provide extensive experiments, ablation studies demonstrating the pertinence of the approach. It will certainly inspire many researchers working on GNNs for protein related tasks and beyond.



**Metareview: Summary, Strengths And Weaknesses:**

The paper proposes a novel method for protein inverse folding, with some strong elements, such as the use of virtual atoms, and the consideration of multi-scale residue interactions. The approach achieves higher recovery than comparison approaches with great efficiency.

The reviewers and AC all agree that the paper makes some very nice novel contributions and that the presentation is clear. The newly added ablation studies significantly improve the significance of this study. We also thank the authors for providing new experimental results on CATH4.3, with and without AF data augmentation.

We urge the authors to make their code public upon acceptance.

Suggestion for future work: Recovery and perplexity are important metrics for protein inverse folding. However, diversity is equally relevant as ultimately, the goal of inverse folding is to design novel sequences that fold into a desired structure, not to merely recover perfectly the native sequence. It would be interesting to therefore assess the diversity of the sequences output by PiFold.

**Note From Pc:**

if the above contains the word "oral" or "spotlight" please see: "oral" presentation means -> notable-top-5% and "spotlight" means -> notable-top-25%. As stated in our emails, we are disassociating presentation type from AC recommendations